# A Novel, Individualized Exercise Program for Patients with Peripheral Arterial Disease Recovering from Bypass Surgery

**DOI:** 10.3390/ijerph16122127

**Published:** 2019-06-16

**Authors:** Edita Jakubsevičienė, Karolina Mėlinytė, Raimondas Kubilius

**Affiliations:** 1Institute of Cardiology, Medical Academy, Lithuanian University of Health Sciences, Sukilėlių St. 15, Kaunas 50103, Lithuania; 2Department of Cardiology, Medical Academy, Lithuanian University of Health Sciences, Eivenių St. 2, Kaunas 50009, Lithuania; karolina.mlinyt652@gmail.com; 3Department of Rehabilitation, Medical Academy, Lithuanian University of Health Sciences, Eivenių St. 2, Kaunas 50009, Lithuania; Raimondas.Kubilius@kaunoklinikos.lt

**Keywords:** quality of life, exercise program, risk factors

## Abstract

The effectiveness of an individual six-month-long physical exercise program in improving health-related quality of life (HRQOL) is unclear. There is some evidence that an individual exercise program can be effective for this aim. The goal of this study was to compare an individual six-month-long physical exercise program for patients with PAD (Peripheral Arterial Disease) with a traditional exercise program and find the effect of these programs on HRQOL and PAD risk factors. The study included patients who underwent femoral–popliteal artery bypass grafting surgery. Patients were divided into three groups: patients participating in an individual six-month-long physical exercise program (group I), in the standard physical activity program (group II), and in a control group (group III), with no subjects participating in rehabilitation II. Results: group I patients had a significantly (*p* < 0.001) higher HRQOL at 6 months after their surgery compared with groups II and III. The HRQOL scores were significantly (*p* < 0.05) lower after surgery among older (≥ 65), overweight participants, as well as among patients with diabetes mellitus and cardiovascular diseases when comparing study results with patients without these risk factors.

## 1. Introduction

Peripheral artery disease (PAD) is a global problem, and it is recognized as an independent risk factor for both myocardial and cerebrovascular events, which are the leading causes of mortality worldwide. Traditional cardiovascular disease risk factors, such as age, diabetes mellitus (DM), smoking, hypertension, dyslipidemia, and physical inactivity, are strongly associated with the development and progression of PAD [1,2]. When PAD is declared as a chronic condition, the lower limb arterial blood flow surgery is a first-line treatment method in a cure-oriented field [3,4]. It is important to notice that structured rehabilitation is recommended for these patients and the basic component of rehabilitation is supervised exercise therapy (a Class IA level of evidence) [5]. Unsupervised exercise therapy is indicated when supervised exercise therapy is not feasible or available (a Class IC level of evidence). Patients who are unable to participate in supervised exercise therapy may choose home-based walking exercise intervention, which is effective in patients with PAD. Patients with PAD have a significantly worse health-related quality of life (HRQOL) than people without this disease [2]. It is associated with developing intermittent claudication and critical limb ischemia [6]. Because of pain, patients often avoid physical activity, especially ambulation, which leads to the additional decline in functional status and HRQOL. Usually, HRQOL is improved after lower limb arterial blood flow surgery and physical load program, which is the basic component of rehabilitation for patients with PAD. However, sometimes it is difficult to define improvement in the functional status of patients with peripheral arterial disease because of the expressed complicated comorbid conditions [7]. Nevertheless, most patients with severe HRQOL impairment before surgery show significant improvement in the quality of life measured several weeks after the intervention [7,8]. One of the main health care goals in PAD after lower-limb bypass surgery is to integrate the use of long-term rehabilitation programs to provide a longer treatment regimen without interruption wherever possible [5]. However, there is still a lack of studies showing the best long-term physical exercise program and proving the effectiveness of rehabilitation after lower-limb arterial blood flow surgery. Not many studies have evaluated the correlation between patient-reported symptoms concerning HRQOL and PAD risk factors following peripheral revascularization and rehabilitation as a means for assessing treatment outcomes. Exercise rehabilitation programs result in significant improvements in walking longer distances and are recommended as first-line therapies, but long-term effects of different rehabilitation programs are uncertain [5]. However, it is important to notice that a structured community- or home-based exercise program is recommended for these patients (a Class IIA level of evidence) [5,9,10]. The aim of this study was to compare an individual six-month-long physical exercise program for patients with PAD with a traditional exercise program and find the effect of these programs on HRQOL and PAD risk factors. The novelty of the study was the fact that the efficacy of an individual six-month-long physical exercise program for patients with PAD was evaluated and the influence of risk factors on HRQOL before and after lower limb bypass surgery was determined.

## 2. Materials and Methods

### 2.1. Study Population

The study population consisted of patients who were hospitalized at Hospital of the Lithuanian University of Health Sciences (LUHS) and Kaunas Clinical Hospital and diagnosed with a superficial femoral artery occlusion. These patients underwent femoral–popliteal artery bypass grafting surgery. Ethical approval was obtained for the study (issued by the Kaunas Region Biomedical Research Ethics Committee of LUHS (No. BE-2-22)), and all participants gave a written informed consent prior to their enrollment.

Exclusion criteria were as follows: critical limb ischemia, stage IV of PAD as defined by Fontaine; limb amputation; patients with neoplastic diseases; presence of dementia or other mental disorder; limited walking due to orthopedic disorders; patients with angina pectoris or other heart diseases; occlusion of other arteries; and patients who did not give their consent to participate in the study.

Inclusion criteria were as follows: diagnosis of a superficial femoral artery occlusion (more than two weeks from the beginning of pain), stage II–III of peripheral arterial disease as defined by Fontaine classification, and femoral–popliteal artery bypass grafting.

All studied patients underwent femoral–popliteal artery bypass grafting. A large subcutaneous vein or vascular prosthesis was used. The upper graft was created in the femoral artery area, and the lower in the popliteal artery P1 segment, where the superficial femoral artery leaves the adductor canal and enters the popliteal fossa.

The study group was randomly assigned into two rehabilitation groups according to the rate 1:1 (considering the inclusion number, equal numbers were attributed to group I and unequal numbers to group II). The first rehabilitation (intervention) group (group I) consisted of patients with an individual six-month-long exercise program. The second rehabilitation (intervention) group (group II) was made up of patients with the standard physical activity program. The subjects of rehabilitation groups I and II who refused stage II rehabilitation formed the third, control group (III).

### 2.2. The Stage of Rehabilitation

The general program of these (I, II, and III) groups in rehabilitation I consisted of combined training. During the first 2 days, static breathing exercises, relaxation, and general light small muscle training exercises were used. During the next 5 days, dynamic breathing exercises, physical exercises for medium sized and large muscle development, and walking therapy were applied. The first stage of rehabilitation consisted of 7 days of treatment; after this, early rehabilitation and optimal medical treatment were started. After hospitalization, patients of groups I and II were referred for stage II rehabilitation. The rehabilitation program performed in rehabilitation centers lasted for 20 days and consisted of pharmacologic treatment, evaluation of risk factors and their modification, supervised exercise therapy twice a day (individual long-term exercise program—group I, the standard physical activity program—group II), occupational therapy, physiotherapy, psychologist’s and social worker’s consultations. The rehabilitation program was carried out six times a week. The head of rehabilitation center and physical therapists were involved in the investigation and took care of the study subjects. Rehabilitation program was designed for every patient individually in accordance with the patient’s needs.

The control group did not participate in rehabilitation centers, but was assigned an individualized physical activity program and continued optimal pharmacological treatment at home under supervision of a family doctor.

### 2.3. Individual Tong-Term Exercise Program (Group I)

An individual long-term exercise program was carried out by the I group for six months. This program was designed according to the recommendations for patients with PAD and after lower limb arterial blood flow surgery [5,11,12,13]. Individual physical therapists worked with the subjects up to 45 min. Procedures consisted of the following: (1) a 5–10 minutes warm-up consisting of dynamic breathing and stretching exercises; (2) lower limb exercising including track walking, stair climbing, and treadmill exercises; (3) a 5–10 minutes cooling down consisting of static and dynamic breathing and stretching exercises. The intensity of training was based on clinical conditions and established between 60% and 85% of maximum heart rate. The rehabilitation program was carried out six times a week. After 20 days of rehabilitation, a group with this program was assigned to an individualized physical activity program and continued optimal pharmacological treatment at home for five months under the supervision of a family doctor. In this program, patients were asked to walk for at least 30 minutes a day three to five times a week and to increase their walking time as often as possible, including a warm-up phase and ending with a cooling down period. In addition, the studied rehabilitees were provided with a physical activity diary that included instruction on how to complete it. In the diary, the study subjects had to indicate the date, duration of, and description of their activity as well as to record their wellbeing.

### 2.4. The Standard Physical Activity *Program* (Group II)

The standard physical activity program was applied to the second (II) group. The second group (II) consisted of patients with the standard physical load program designed for cardiovascular diseases patients. This program consisted of combined aerobic and resistance training. It was performed using treadmills, ergometers, steppers, stair climbing, and jogging and resistance devices in the rehabilitation centers under the supervision of a physical therapist. Each session lasted up to 60 minutes, including a 10-minute warm-up followed by 40 minutes of aerobic and resistance training, and finally, a 10-minute cooling down. The intensity of training was based on clinical conditions and established between 60% and 85% of maximum heart rate. The rehabilitation program was carried out six times a week. After rehabilitation treatment, the subjects were recommended to continue physical training programs and optimal pharmacological treatment under the supervision of a family doctor.

### 2.5. Patients Data Collection

Patient data were collected from case histories. The analysis of the patient data comprised of the following factors: gender, age, smoking, diabetes mellitus, hypertension, and cardiovascular disease (the risk factors were based on medical history and diagnosed by a physician).

HRQOL was evaluated using the SF-36 questionnaire, which is one of the most commonly used instruments and is applicable in many health fields. This questionnaire consisted of eight parts (physical, role and social functioning, mental health, patient health perceptions, vitality, bodily pain, and change in health) that were measured. In this questionnaire, each individual domain was transformed to a score from 0 to 100, with 100 being the best possible result. Results were divided into physical and mental component summary scores. The SF-36 questionnaire is a practical tool for routine assessment of patient outcomes in medical practice as well as for research [2,6].

### 2.6. Statistical Analysis

The statistical analysis of our study was performed using the SPSS for Windows 21.0 program (IBM, Armonk, NY, USA). The quantitative variables of groups were compared using student’s (t) test, and were presented as the arithmetic mean and standard error of the mean value (SE). Qualitative determinants were expressed as a percentage; their interdependence was assessed using Chi-square test (χ^2^). Independent two-sample t-test was used, and u-test was used for small samples. T-test for pair samples was used to compare repetitive measurements. The treatment effect after 6 months was assessed using analysis of covariance (ANCOVA). In the ANCOVA, the data after 6 months were used as dependent variables, the variables before the operation as a covariate, and the patients group as a factor. For the comparison of two samples, dispersive analysis was used. Verification of statistical hypotheses selected statistical confidence level of *p* < 0.05 (95% statistical confidence).

## 3. Results

The study included 218 patients who were diagnosed with the superficial femoral artery occlusion and underwent femoral-popliteal artery bypass grafting surgery. Out of 218 studied patients, 58 failed to complete the full study follow-up period: 25 patients of the first group, 23 patients of the second group, and 10 patients of the third group. Of those patients, four patients were re-operated on, four patients had amputation above the knee, one patient had amputation of two toes, twenty patients were unable to continue the study due to health problems (cardiovascular diseases, oncological diseases, and other chronic diseases), and twenty-nine patients were lost for follow-up examination (failed to contact or refused to participate). The data of these patients were not included in the analysis (Table 1).

General demographic and clinical characteristics of all study participants are summarized in Table 2. 160 patients completed our research: the first group consisted of 63 patients, the second group of 65 patients, and the control group of 32 patients.

The mean age of patients who participated in the study was 67.9 ± 1.1 years. All the study groups contained participants of similar age at baseline. There was no significant difference (*p* > 0.05) with respect to body mass index (BMI), smoking, diabetes mellitus (DM), cardiovascular diseases (CVD), and hypertension among all three study groups. Among the subjects, 93 % were men and 70 % were smokers. Blood tests for total cholesterol, low- and high-density lipoprotein cholesterol, and triglyceride were similar among all groups (*p* > 0.05). Obesity (BMI > 25 kg/m^2^) was observed among all subjects. 65.6 % of subjects had hypertension, only 7.5 % had DM, and 51.3 % of the study subjects had CVD (stroke, myocardial infarction, heart failure, angina pectoris, and coronary artery surgery).

### 3.1. Quality of Life Before Surgery

Measurement of SF-36 questionnaire points obtained before surgery showed that the fullness of life of group subjects was assessed similarly: there was no significant difference from the point of view of physical and mental health (*p* > 0.05). Quality of life in accordance with SF-36 questionnaire is provided in Table 3.

Significant differences were not noticed while analyzing data of intervention groups I and II. However, the analysis of quality of life in intervention group I and control group III shows a significant difference of physical activity (*p* < 0.05). Physical activity was assessed better for subjects of intervention group I by 33 points on average, and subjects of control group by 28 points on average. Data of quality of life in accordance with SF-36 questionnaire is provided in Table 4.

### 3.2. Quality of Life Changes at 6-month Follow-Up

Analysis of the quality of life changes at 6-month follow-up showed that they were statistically significantly higher in intervention group I considering aspects of physical health, physical activity, bodily pain, mental health, social function, and general life quality (*p* < 0.05). Data on change of the quality of life of intervention groups I and II have been analyzed. It was found out that at 6-month follow-up changes among intervention group I appeared in five areas: scores of physical health, physical activity, bodily pain (i.e., bodily pain decreased), social function, and vitality statistically significantly (*p* < 0.05) improved and general life quality was also significantly higher (*p* < 0.001). Changes of the quality of life at 6-month follow-up in accordance with SF-36 questionnaire are provided in Table 5.

### 3.3. Changes of Quality of Life Considering Risk Factors

Analysis of the HRQOL data changes of all study participants (I, II, and III groups) after 6 months after surgical intervention has revealed that social functions improved more (*p* < 0.05) in younger patients (< 65 years) and in those with BMI < 30. Additionally, limitation of activity due to physical dysfunction improved (*p* < 0.05) among patients with BMI < 30. Study subjects whose anamnesis showed no other CVD experienced statistically significant reduction of bodily pain (*p* < 0.05). General health improved (*p* < 0.05) in people without DM, and vitality was bigger (*p* < 0.05) among younger patients (< 65 years). Changes of physical and mental health indices after 6 months with respect to risk factors are presented in Table 6 and Table 7.

Summarizing the data of our study, we can state that improvement of HRQOL after lower limb artery bypass surgery is evident to all study participants irrespective of various risk factors. However, after analyzing HRQOL changes more attentively we have found out that such risk factors as age (≥ 65years), overweight (BMI ≥ 30), DM, and CVD have a negative influence on the improvement of HRQOL. 

## 4. Discussion

PAD is a common disease that is associated with cardiovascular mortality and worse functioning in daily life [14,15]. The rehabilitation system for PAD patients was developed in our country more than 20 years ago. However, attention has been paid exclusively to patients who have received reconstructive surgery of coronary arteries and those who have suffered myocardial infarction. Lower limb reconstructive surgery still is a commonly used method of treatment of PAD, and we looked at exactly such patients in our study. 

Most international authors believe that physical activity is useful for patients with PAD [13]. This is related to the improvement of autonomic functions, acceleration of lipid metabolism, decreased obesity risk, normalization of arterial blood pressure, reduction of inflammatory processes, and increased skeletal muscle glucose uptake. There are also other authors who have also discussed the effectiveness of physical activity in patients with post-lower-limb bypass surgery [16,17,18,19].

One of the biggest advantages of our study is assessment of the effectiveness of an individual 6-month-long exercise program in patients after lower limb bypass surgery. As a result of our study, the effect of this program on peripheral circulation, functional status recovery, and HRQOL was established. The basic risk factors that had influence on HRQOL were also appreciated. In most other studies, the program of physical activity is compared with endovascular treatment [20,21,22]. In the management of patients with PAD, lower limb reconstructive surgery does not provide significantly bigger benefits compared with rehabilitation exercises alone in terms of improvement in functional performance or HRQOL [23]. To assess the efficiency of the 6-month-long exercise program, the HRQOL in patients with PAD was studied. The SF-36 questionnaire provides useful information about the impact of PAD on the patients’ lives. It is a tool to assess HRQOL and many authors used this method to estimate the efficiency of patient treatment [2,6,24]. The HRQOL of patients with PAD worsens each year. The patients with this disease are socially inactive and dependent on others. To assess the efficiency of physical program for HRQOL, the data between and of all three groups were compared. It was noticed that the physical and mental health in both intervention groups was distributed similarly (*p* = 0.035). However, our study results showed that the first group with a six-month-long exercise program had better results in the physical and mental health component of HRQOL in comparison with the control group. Bo et al. reported better results of the SF-36 in the physical part (*p* = 0.004) in the intervention group of hospital-based supervised exercise therapy [22]. A recent meta-analysis has shown that walking training to various levels of claudication pain improves perceived walking as measured by self-reported physical function (SF-36) in people with PAD. [25]. Physical, emotional, and mental health is affected by PAD. The study by Wu A et al. showed that the association with lower ankle-brachial index was more evident for physical components of the quality of life than for mental components. However, it is important to emphasize that the SF-12 was used in their study [6]. The evaluation of HRQOL is similar in patients with PAD and with another CVD [26].

The HRQOL was verified to consider risk factors. Age was the most powerful influence for HRQOL. It was noticed that patients younger than 65 years were more physically active, viable, and socially active than older patients. The impact of age on the quality of life was established many years ago [27]. In our study, it was established that BMI ≥ 30 kg/m^2^ had a negative influence on daily activity and social functions. Physical activity also decreased in patients with DM. A negative impact of the described risk factors on HRQOL in patients with PAD is considered to be most important not only in our study, but also in many other scientific information sources [23,25,28,29]. The HRQOL was assessed 6 months after surgery. This assessment revealed that statistically more significant changes were in the six-month-long physical exercise program in the spheres of physical and mental health. International authors presented ambivalent results. Bo et al. reported that a 6-month-long physical activity program for patients after lower limb arterial blood flow surgery had a positive influence on HRQOL (*p* = 0.004) [21]. However, Badger et al. stated that additional physical exercise programs had no influence on the HRQOL of patients after leg artery bypass surgery [16]. Mazari et al. spoke of similar results: after percutaneous transluminal angioplasty the HRQOL does not change irrespective of the application or not of a physical exercise program [30]. All the authors mentioned above used the SF-36 questionnaire to evaluate the HRQOL.

Aherne et al. examined compliance amongst patients attending a community-based supervised exercise therapy program in their study and elucidated the factors affecting compliance and symptomatic improvement. The results of their research showed that active smokers with hypercholesterolemia, active smokers with ischemic heart disease, and older people were complying less in comparison with their counterparts [27]. Similarly to our study, Oakley et al.’s research results revealed that a 12-week home exercise program had a significant influence on improving patients’ walking distance and walking speed compared with normal walking. The HRQOL and ankle-brachial pressure indices improved only after 3 months, and it is suggested that this improvement was caused by collateral arterial development [31]. In Kruidenier et al.’s research, physical and mental health of the patients 6 months after lower limb arterial blood flow surgery [32] was lower than similar heath indices in the first interventional group of our study. It is important to notice that in Kruidenier et al.’s study, the program of physical activity was started 3 weeks after surgery. In our study this program was started on the first day after reconstructive surgery.

There are several limitations to this study. First, only patients with femoral-popliteal arterial disease were included; thus, a large number of patients with a bilateral or mixed arterial disease were excluded. However, this was necessary from the scientific point of view in order to reduce the influence of confounders and to provide a robust answer to the treatment controversy in this group. Second, the subjects of rehabilitation groups I and II who refused stage II rehabilitation formed the third group—control group (III). The control group did not participate in rehabilitation centers, but was assigned an individualized physical activity program and continued optimal pharmacological treatment at home under supervision of a family doctor.

## 5. Conclusions

Assessing changes in the HRQOL six months after surgery, we have found out that statistically more significant changes occured in an individual six-month-long physical exercise program in the spheres of physical and mental health. Thus, increased physical capacity of the patients improved the HRQOL with respect to physical health: study subjects became more active physically, more energetic, and more self-reliant. They had a better opinion about their general health, and their social functions and emotional condition improved as well. Therefore, we can postulate that the improvement of the patients’ functional status and quality of life also improved. Having analysed the changes of the HRQOL in terms of risk factors 6 months after surgical intervention, we noticed that age (≥65), overweight (BMI ≥ 30), DM, and CVD had negative impacts on the improvement of the HRQOL.

To summarize our study, we have emphasized the usefulness of a 6-month-long individual physical load program after performance of lower limb arterial blood flow surgery as an important factor for better HRQOL.

## Figures and Tables

**Table 1 ijerph-16-02127-t001:** Distribution in groups of patients who did not complete the study according to reasons.

Reasons	Group I (n = 88)	Group II (n = 88)	Group III (n = 42)
Re-operated, n (%)	2 (2)	2 (2)	–
Leg amputation above knee, n (%)	1 (1)	2 (2)	1 (2)
Toe amputation, n (%)	1 (1)	–	–
Health problems, n (%)	7 (8)	9 (10)	4 (10)
Did not come, n (%)	14 (16)	10 (11)	5 (12)
Total, n (%)	25 (28)	23 (26)	10 (24)

Notes: Group I = patients with a 6-month-long exercise program; Group II = patients with the standard physical load program; and Group III = control group; n = number, percentage presented of number (n) of studied persons in a group.

**Table 2 ijerph-16-02127-t002:** General demographic and clinical characteristics.

Characteristics	Group I (n = 63)	Group II (n = 65)	Group III (n = 32)	*p*-Value	Total (n = 160)
Age, V (± SE)	68.2 (1.0)	67.3 (0.9)	68.3 (1.4)	0.322	67.9 (1.1)
*Gender*					
Women, n (%) Men, n (%)	5 (7.9) 58 (92.1)	4 (6.2) 61 (93.8)	2 (6.3) 30 (93.8)	0.912	11 (6.9) 149 (93.1)
BMI kg/m2, V (± SE)	26.4 (0.4)	27.2 (0.5)	26.6 (0.6)	0.461	26.8 (0.3)
*Smoking* Smoker, n (%) Non-smoker, n (%)	47 (74.6) 16 (25.4)	45 (69.2) 18 (30.8)	20 (62.5) 12 (37.5)	0.371	112 (70) 48 (30)
*Blood tests* TC (mmol/l), V (± SE) LDL-C (mmol/l), V (± SE) HDL-C (mmol/l), V (± SE) TG (mmol/l), V (± SE)	4.8 (0.1) 3.0 (0.1) 1.1 (0.1) 1.7 (0.1)	5.1 (0.1) 3.2 (0.1) 1.1 (0.1) 1.8 (0.1)	4.7 (0.2) 2.9 (0.1) 1.1 (0.1) 1.7 (0.1)	0.209 0.268 0.309 0.576	4.9 (0.1) 3.1 (0.1) 1.1 (0.1) 1.7 (0.1)
*Comorbidities*					
DM, n (%)	5 (7.9)	5 (7.7)	3 (6.3)	0.955	13 (7.5)
CVD, n (%)	33 (52.4)	31 (47.7)	18 (56.3)	0.711	82 (51.3)
Hypertension, n (%)	41 (65.1)	45 (69.2)	19 (59.4)	0.626	105 (65.6)

Notes: Group I = patients with a 6-month-long exercise program; Group II = patients with the standard physical load program; Group III = control group; V = mean; SE = standard error; n = number; percentage presented of number (n) of studied persons in a group; BMI = body mass index; TC = total cholesterol level; LDL-C = low-density lipoprotein cholesterol level; HDL-C = high-density lipoprotein cholesterol level; TG = triglyceride level; DM = diabetes mellitus; CVD = cardiovascular disease.

**Table 3 ijerph-16-02127-t003:** Data of quality of life of subjects before surgery.

Quality of Life Aspects	Group I (n = 63) V (SE)	Group II (n = 65) V (SE)	Group III (n = 32) V (SE)	*p*-Value among Groups	Total (n = 160) V (SE)
Physical health	33.11 (0.8)	33.35 (0.9)	32.41 (1.1)	0.823	33.07 (0.5)
Physical activity	33.80 (1.5)	33.92 (1.5)	28.43 (1.5)	0.079	32.78 (0.9)
Role limitation	32.93 (1.7)	34.61 (2.0)	34.37 (3.3)	0.825	33.90 (1.2)
Bodily pain	28.57 (1.8)	29.57 (1.7)	29.51 (2.2)	0.910	29.16 (1.1)
General health score	37.14 (1.4)	35.30 (1.2)	37.34 (2.0)	0.551	36.43 (0.8)
Mental health	47.82 (1.2)	45.12 (1.2)	45.41 (1.6)	0.256	46.24 (0.7)
Role limitation	39.68 (1.9)	38.23 (1.7)	37.18 (2.0)	0.698	38.59 (1.1)
Social function	38.27 (1.7)	36.92 (1.3)	37.15 (2.1)	0.818	37.50 (0.9)
Emotional state	52.90 (2.7)	50.25 (1.4)	52.08 (5.7)	0.872	51.66 (2.2)
Vitality	59.44 (1.7)	55.07 (1.4)	55.25 (1.7)	0.085	57.22 (0.9)
General life quality	40.47 (0.7)	39.23 (0.6)	38.91 (1.1)	0.372	39.65 (0.4)

Notes: Group I = patients with a 6-month-long exercise program; Group II = patients with the standard physical load program; Group III = control group; V = mean; SE = standard error; n = number; percentage presented of number (n) of studied persons in a group.

**Table 4 ijerph-16-02127-t004:** Comparison of subjects’ quality of life among groups before surgery.

Quality of Life Aspect	*p*-Value between Group I and II	*p*-Value between Group I and III
Physical health	0.848	0.635
Physical activity	0.960	0.035
Role limitation	0.535	0.676
Bodily pain	0.692	0.755
General health score	0.331	0.935
Mental health	0.121	0.251
Role limitation	0.581	0.425
Social function	0.546	0.703
Emotional state	0.594	0.884
Vitality	0.063	0.064
General life quality	0.236	0.255

Notes: Group I = patients with a 6-month-long exercise program; Group II = patients with the standard physical load program; Group III = control group.

**Table 5 ijerph-16-02127-t005:** Quality of life changes of subjects after 6 months.

Quality of Life Aspect	Group I *∆* (SE) (n = 63)	Group II *∆* (SE) (n = 65)	Group III *∆* (SE) (n = 32)	*p*-Value among Groups	*p*-Value between Group I and II
Physical health	25.82 (1.7)	15.66 (1.9)	14.40 (2.2)	<0.001	<0.001
Physical activity	35.15 (2.1)	16.46 (2.6)	21.87 (3.2)	<0.001	<0.001
Role limitation	18.01 (4.6)	8.07 (4.8)	7.81 (5.8)	0.251	0.141
Bodily pain	39.50 (2.8)	23.41 (3.1)	23.26 (4.0)	<0.001	<0.001
General health score	11.34 (2.1)	8.38 (2.0)	4.68 (15.7)	0.178	0.321
Mental health	12.18 (2.1)	9.54 (1.9)	11.96 (1.9)	0.008	0.367
Role limitation	19.57 (2.7)	14.87 (4.1)	10.41 (6.0)	0.345	0.350
Social function	32.27 (2.7)	24.10 (2.8)	20.48 (3.3)	0.020	0.035
Emotional state	6.53 (1.9)	8.92 (2.0)	7.75 (3.1)	0.721	0.409
Vitality	18.33 (3.0)	10.30 (2.6)	9.21 (3.1)	0.064	0.050
General life quality	22.59 (1.0)	14.31 (1.1)	13.18 (1.3)	<0.001	<0.001

Notes: Group I = patients with a 6-month-long exercise program; Group II = patients with the standard physical load program; Group III = control group; ***∆*** = change; SE = standard error; n = number; percentage presented of number (n) of studied persons in a group.

**Table 6 ijerph-16-02127-t006:** Changes of the means of physical health indices of study subjects after 6 months with respect to risk factors.

Risk Factors	Physical Health	Physical Activity	Role Limitation of Physical Health	Bodily Pain	General Health
	*∆* (SE)	*∆* (SE)	*∆* (SE)	*∆* (SE)	*∆* (SE)
Age					
≥65	17.71 (2.2)	24.30 (2.0)	11.89 (3.6)	28.41 (3.4)	7.7 (2.4)
<65	22.20 (1.7)	26.15 (2.7)	12.01 (4.9)	30.34 (2.4)	9.30 (1.5)
Gender					
Men	21.15 (1.4)	24.96 (1.6)	13.15 (3.0)	30.12 (2.0)	8.99 (1.3)
Women	14.18 (4.2)	24.09 (8.2)	4.54 (10.5)	24.24 (8.0)	6.36 (3.7)
BMI, kg/m^2^					
≥30	16.70 (3.0)	23.00 (3.8)	2.14 (6.4)	25.07 (4.6)	9.00 (2.8)
<30	21.91 (1.5)	25.44 (1.8)	14.68 (3.2) *	31.02 (2.1)	8.76 (1.4)
Smoking					
Yes	19.03 (1.3)	24.96 (1.6)	12.80 (2.9)	29.53 (2.0)	8.5 (1.3)
No	15.09 (5.9)	23.00 (7.1)	6.80 (25.7)	35.55 (7.3)	16.00 (5.7)
Hypertension					
Yes	19.68 (1.51)	26.57 (2.0)	9.38 (3.6)	29.41 (2.4)	10.04 (1.6)
No	23.44 (3.1)	21.72 (2.6)	16.81 (4.9)	30.30 (3.2)	6.45 (2.3)
DM					
Yes	19.19 (4.9)	17.91 (7.1)	4.08 (9.9)	24.07 (9.5)	8.10 (1.3)
No	20.79 (1.4)	25.47 (1.6)	10.73 (3.0)	30.18 (2.0)	17.50(4.9) *
CVD					
Yes	21.05 (1.9)	25.36 (2.1)	13.41 (4.0)	26.28 (3.0) *	9.75 (1.9)
No	20.27 (1.9)	24.42 (2.4)	10.38 (4.2)	33.33 (2.4)	7.82 (1.7)

Notes: ∆ = change; SE = standard error; BMI = body mass index; DM = diabetes mellitus; CVD = cardiovascular disease; * = *p* < 0.05.

**Table 7 ijerph-16-02127-t007:** Changes of the means of mental health indices of study subjects after 6 months with respect to risk factors.

Risk Factors	Mental Health	Role Limitation of Mental Health	Social Function	Emotional State	Vitality
	*∆* (SE)	*∆* (SE)	*∆* (SE)	*∆* (SE)	*∆* (SE)
Age					
≥ 65	10.28 (1.7)	14.19 (2.6)	20.72 (2.7)	6.92 (1.5)	7.50 (2.8)
< 65	11.91 (2.6)	19.23 (4.6)	29.42 (2.1) *	9.46 (2.3)	16.01(2.1) *
Gender					
Men	10.56 (1.5)	16.33 (2.4)	26.47 (1.7)	7.51 (1.3)	12.78 (1.7)
Women	14.18 (4.2)	9.09 (7.9)	28.28 (6.5)	10.90 (5.6)	19.54 (9.4)
BMI, kg/^2^					
≥ 30	10.12 (1.7)	14.40 (2.5)	24.17 (1,8)	8.00 (2.3)	15.14 (3.8)
< 30	13.10 (2.3)	20.95 (5.4)	35.23 (3.6) **	7.68 (1.5)	12.72 (1.9)
Smoking					
Yes	10.74 (1.4)	16.12 (2.3)	26.73 (1.7)	7.50 (1.2)	12.96 (1.7)
No	16.76 (3.5)	6.66 (12.4)	22.21 (9.2)	15.20 (12.8)	22.00 (8.8)
Hypertension					
Yes	12.22 (1.6)	15.56 (3.0)	28.14 (2.0)	8.38 (1.5)	12.47 (2.1)
No	6.88 (3.1)	16.36 (3.6)	23.63 (3.0)	6.54 (2.3)	14.72 (3.0)
DM					
Yes	13.68 (3.9)	16.66 (8.7)	29.62 (6.8)	2.00 (3.4)	7.50 (7.2)
No	10.57 (1.5)	15.76 (2.4)	26.35 (1.7)	8.21 (1.3)	13.71 (1.8)
CVD					
Yes	10.62 (2.1)	12.60 (2.9)	26.96 (2.1)	8.19 (1.8)	11.89 (2.4)
No	11.05 (2.0)	19.23 (3.6)	24.21 (2.6)	7.2 (1.8)	14.67 (2.5)

Notes: ∆ = change; ± SE = standard error; BMI = body mass index; DM = diabetes mellitus; CVD = cardiovascular disease; * *p* < 0.05; ** = *p* < 0.01.

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
