# Peer review of "A Novel, Individualized Exercise Program for Patients with Peripheral Arterial Disease Recovering from Bypass Surgery"

_ijerph, 2019, doi:10.3390/ijerph16122127_

Round 1
Reviewer 1 Report
Thank you for taking the time to complete this extensive piece of work. I am aware that this would have taken a considerable amount of time. I query about the novelty and originality of this research. There is a wealth of literature out on pub med stating that supervised rehabilitation if advisable for individuals with Peripheral arterial disease and also how this affects quality of life. This article supports those findings, but it is not original enough to support publication.
Author Response
We included more recent studies on this topic, the novelty of the study and several limitation of this study.
Line 2 corrected to Exercise;
Line 14, 17, 21, 296, 297, 304, 322 corrected to individual six-month-long physical exercise program;
Line 28 included point and comma after each word;
Line 34, 94, 98, 194, 236, 273 corrected grammar of the words;
Line 42, 44 corrected to much space between phrases;
Line 35, 38, 105, 279 included more recent studies on this topic;
Line 56-62 included the aim of the research and the novelty of the study;
Line 359-366 included the limits of the research;
Line 15, 326, 330, 348 - chanced to abbreviations.

Reviewer 2 Report
Please detailed why did that ''The effectiveness of rehabilitation ......is unclear''.....I think the individual exercise help always for this aim.
Line 27 - please include point and comma after each word.
Lines 41, 43 - to much space between phrases, please arrange.
Please include in Introduction more recent studies on this topic.....also, please include the aim of the research and the novelty of the study.
Please detailed why the control group did not participate in rehabilitation? which was the causes? how they are evaluated? because in results i see many dates.....please detailed
Please include which is the limits of the research.
Author Response
Line 2 corrected to Exercise;
Line 14, 17, 21, 296, 297, 304, 322 corrected according to your recommendations;
Line 28 included point and comma after each word;
Line 34, 94, 98, 194, 236, 273 corrected grammar of the words;
Line 42, 44 corrected to much space between phrases;
Line 35, 38, 105, 279 included more recent studies on this topic;
Line 56-62 included the aim of the research and the novelty of the study;
The control group did not participate in rehabilitation because the subjects of rehabilitation groups I and II who refused (did not want to participate in rehabilitation) stage II rehabilitation formed the third, control group (III). We included in the limitation of this study - Line 364-368;
Line 359-366 included the limits of the research;
Line 15, 326, 330, 348 - chanced to abbreviations.

Round 2
Reviewer 2 Report
Please arrange the space between tables.
Author Response
Introduction – corrected and emphasized further the uniqueness of the study: line 51-55
This trial was randomized and controlled trial (line 86). The trial was registered by the Kaunas Region Biomedical Research Ethics Committee of LUHS (No. BE-2-22) (line 72-74).
Line 88, 92, 107, 102, 103, 108-110 was corrected and clarified this section
Line 113 the sentence was corrected.
Line 121, 137 “After 20 days of rehabilitation…”yes, 6 days per week as indicated in the previous paragraph. Included the sentence.
Line 131-132 the sentence was corrected.
Individual long-term exercise program (Group I) - this program was designed according to the recommendations for patients with PAD and after lower limb arterial blood flow surgery. The standard physical activity program (Group II) – patients with the standard physical load program designed for cardiovascular diseases patients. Included the sentence - line 132-133.
The hypotheses of the study: individual long-term exercise program is effective for patients who had lower limb bypass surgery. The main risk factors of peripheral artery disease are associated with recovery functional status and quality of life.
“The validity of the questionnaires was tested by conducting an experimental study involving 15 patients.” This sentence was deleted line 143-146.
Line 159, 163-166 was described which tests were done for which variables.
Line 188-189, 192 included p value.
Table 3, table 4 physical fitness chanced to physical activity and fullness chanced to quality
We didn’t comparison in group II and III, because evaluated the effect of individual long-term exercise program on patients with peripheral arterial disease after lower limb bypass surgery
“Gender statistically did not have significant impact for fullness of life: before surgery men and women assessed their health equally” and in the table there is no significant differences.
In table 5 p values was determined T-tests.
Tables 7, 8, 12, 13 were confusing, so we decided deleted items, because this tables just confirmed information which was before provided.
Line 367-369 deleted.
All the text language was modification and changes marked in red color.
Limitations integrated after introduction.
